# Randomised Controlled Trial of Nutritional Supplement on Bone Turnover Markers in Indian Premenopausal Women

**DOI:** 10.3390/nu13020364

**Published:** 2021-01-26

**Authors:** Pramod B. Umarji, Pankaj Verma, Vivek Garg, Marian Schini, Richard Eastell

**Affiliations:** 1Umarji Healthcare, Pune, Maharashtra 411045, India; pramod.umarji@gmail.com; 2Hindustan Unilever Limited R&D, Gurugram, Haryana 122002, India; pankaj.verma@unilever.com (P.V.); vivek.garg@unilever.com (V.G.); 3Academic Unit of Bone Metabolism, University of Sheffield, Sheffield S10 2NR, UK; r.eastell@sheffield.ac.uk

**Keywords:** osteoporosis, bone turnover markers, vitamin D, vitamin K, vitamin B6, vitamin B12, folate

## Abstract

Young Indian women may be at risk of poor bone health due to malnutrition. The aim of this study was to examine the effects on bone metabolism of a nutritional supplement in women aged 25 to 44. The nutritional supplement was a protein-rich beverage powder fortified with multi-micronutrients including calcium (600 mg), vitamin D (400 IU), and vitamin K (55 mcg) per daily serving, while a placebo supplement was low-protein non-fortified isocaloric beverage powder. This 6-month randomised, controlled trial showed favorable changes in bone turnover markers (decreased) and calcium homeostasis; such changes in older adults have been associated with slowing of bone loss and reduced fracture risk. For example, serum CTX decreased by about 30% and PINP by about 20% as a result of the increase in calcium intake. There were also changes in the ratio of carboxylated to undercarboxylated osteocalcin and such changes have been linked to a slowing of bone loss in older subjects. For example, the ratio increased by about 60% after 3 months as a result in the improvement in vitamin K status. Finally, there were improvements in the status of B vitamins, and such changes have been associated with reductions in homocysteine, but it is uncertain whether this would affect fracture risk. The product was generally well tolerated. This study shows the nutritional supplement holds promise for improved bone health among young Indian women.

## 1. Introduction

Young adults achieve peak bone mass around the age of 30 years. Adequate nutrition is thought to be critical in attaining this. Young Indian women may suffer malnutrition and therefore attain an inadequate peak bone mass. For example, they have a low dietary intake of calcium and poor vitamin D status [1,2]. The consequence of these deficiencies is likely to be low bone mineral density [3], and this may predispose to fracture risk in later life. It is estimated that in India, 50 million people have either osteoporosis of low bone mass [3] and among young Indian women 44% have osteopenia [3].

Bone turnover markers (BTM) may be used to assess bone health, and they are particularly useful in clinical trials [4]; in the presence of low dietary calcium and vitamin D, these markers are usually elevated. High levels of BTM are associated with higher fracture risk in the older woman [5]. There is evidence that these markers may be higher in Indians than in Western populations. The bone turnover markers recommended as reference markers by the International Osteoporosis Foundation are PINP and CTX [6].

Clinical trials of calcium and vitamin D supplementation in young women from a number of countries have shown that there is a consequent decrease in bone turnover markers [7,8,9]. Similar trials on older women have shown benefits for fracture risk [10].

Micronutrients, including selenium, zinc, vitamin B12, vitamin K, and folic acid, may play a role in the bone health of young women. Thus, a nutrient supplement that contained all these components might result in an improvement in bone health. Vitamin K increases the ratio of carboxylated to undercarboxylated osteocalcin, and it also increased the amount of protein specific to bone [11,12,13].

The aims of this study were to evaluate the nutritional status of young Indian women by their dietary record and biochemical assessments and to examine the effect of a nutritional supplement in a randomised controlled trial. The key conclusions were that young Indian women are deficient in certain nutrients. A nutrient supplement improves bone turnover and micronutrient status in some but not all nutrients tested in young Indian women.

## 2. Materials and Methods

### 2.1. Design

This was a double-blind, randomised, single-centre, parallel group, controlled study, conducted between May 2017 and January 2018 at B. J. Medical College and Sassoon Hospital, Pune, India (ClinicalTrials.gov identifier: NCT03155269; study number: 207192). Eligible subjects (25 to 45-year-old premenopausal Indian women) were randomly (1:1) allocated to either the test group (protein-rich beverage powder fortified with multi-micronutrients (MMN)) or the control group (low-protein non-fortified isocaloric beverage powder). The composition of the two products can be seen in Table 1.

At the screening visit (Visit 1), subjects provided their written informed consent to participate in the study. Demographics, medical history, current/concomitant medications, general physical examinations, and vital signs were recorded. Height, weight, and body mass index (BMI) were assessed using anthropometric measurements. Urine pregnancy testing was carried out on all female subjects who were of childbearing potential. After that, the subject’s eligibility was determined based on inclusion/exclusion criteria. Eligible subjects were given a diary and weighing scale to record 7-day food intake.

Eligible subjects attended the clinical study site for the baseline visit (Visit 2: 7 to 10 days after the screening visit) and were asked to bring the completed 7-day diary for assessment of dietary intake of calcium, protein, and other key micronutrients. At the baseline visit, subjects were randomly allocated to receive either the test or the reference product at a 1:1 ratio, which was stratified by age (Stratum 1: Age ≥25 years to <35 years; Stratum 2: Age ≥35 years to ≤45 years). The study centre aimed to recruit a target ratio of 50% of subjects in the 25 to 35 years age band and 50% in the 35 to 45 years age band with a minimum of 40% in either age band. The study product was dispensed per the randomisation schedule, with the first dose being taken (orally) under the observation of site staff.

The randomisation schedule was generated by the Biostatistics Department, GSK CH, prior to the start of the study, using validated internal software (SAS, Marlow, UK). Subjects were instructed to take the study product (30 g of powder made up in 200 mL water), twice daily (morning and preferably in the evening) for a total of six months. Subjects attended on-site visits at baseline and at four follow-up visits; visit 3: 1.5 months (45 ± 7 days from baseline), visit 4: 3 months (90 ± 7 days), visit 5: 4.5 months (135 ± 7 days), and visit 6: 6 months (end of treatment: 180 days ± 7 days). Subjects were fasting at baseline, month 3, and month 6 visits, and blood (12-h fasting) and spot urine samples were collected at baseline, 3 months, and 6 months.

The study product was consumed under the observation of site staff during each visit except for visit 1 and visit 6. The last dose had to be consumed the evening before the month 6 visit. In between study visits, the subjects consumed the study product at home and completed a product compliance diary. Random home visits were made by site staff to ensure compliance of product consumption at home. At each visit, subjects were asked to return sachets of study product (empty and unconsumed). Subjects were considered as non-compliant if they consumed less than 80% of the total amount of the recommended dose (for the entire study duration).

The study protocol was approved by the Institutional Ethics Committee. The study was conducted in accordance with the ethical principles that are outlined in the Declaration of Helsinki, the International Council for Harmonisation of Technical Requirements for Registration of Pharmaceuticals for Human Use (ICH) and all applicable local Good Clinical Practice (GCP) and regulations. Written informed consent was obtained from each participant prior to the performance of any study-specific procedures.

### 2.2. Study Participants

Women aged 25 to 45 years (inclusive) having good general and mental health with no clinically significant and relevant abnormalities in medical history or upon physical examination, and BMI between 18 and 30 kg/m^2^ (inclusive), were included in the study. Women who had attained physiological menopause defined as those without a menstrual period for 12 consecutive months, who were pregnant or were intending to become pregnant over the duration of the study, and women who were breastfeeding were excluded from the study. Women with current or regular use of any prescription, over-the-counter vitamin supplements, or herbal medicine, treatment with bisphosphonates (any dose within the previous 2 years) or other medications known to affect bone (within the previous 6 months); history of metabolic bone disease; any hormonal disorders or disturbances; and bone fracture in last 12 months and those who were having more than two units of alcohol per day and/or were smoking were also excluded. Additionally, women who took any other health food drinks/beverages or supplements or had been on supplements within a month prior to study start, and women who used any medication that was known to influence bone mass were excluded from the study. The use of calcium, vitamin D, and multivitamin supplements on a regular basis were to be stopped 2 months before the onset of the trial.

### 2.3. Endpoints

The co-primary outcome variables were the changes from baseline after 6 months of treatment in the bone resorption marker serum C-terminal cross-linking telopeptide of type I collagen (s-CTX-I) and the bone formation marker expressed as ratio of carboxylated osteocalcin to under carboxylated-osteocalcin (c-OC/uc-OC).

The secondary outcome variables were change from baseline after 3 and 6 months of treatment in bone formation markers: serum N-terminal propeptide of type I procollagen (s-PINP), bone alkaline phosphatase (BSAP), serum alkaline phosphatase (s-ALP) and c-OC/uc-OC (at 3 months); change from baseline after 3 and 6 months of treatment in bone resorption markers urinary CTX-1, serum N-terminal telopeptide of type 1 collagen (s-NTX-1), and serum CTX-1 (at 3 months); change from baseline after 3 and 6 months of treatment in calcium status determined by serum calcium, parathyroid hormone (s-PTH), and urinary calcium to creatinine ratio (Ca/CR). Finally, changes from baseline after 3 and 6 months of treatment in other bone metabolism parameters and micronutrients including phosphorus, selenium, vitamin D, folate, vitamin B12, vitamin B6, and zinc were assessed.

Safety assessments included assessment of adverse events (AEs) and serious AEs (SAEs). Other safety assessments included laboratory tests, vital signs, physical examination, and haemoglobin examination.

### 2.4. Laboratory Methods

The following laboratory tests were performed by enzyme-linked immunosorbent assays: c-OC, ucOC (Takara Bio Inc, Shiga, Japan) and serum CTX (Immunodiagnostic Systems, Boldon, UK), BSAP, serum folate, and vitamin B12. Calcium was measured by spectrophotometry. Electrochemiluminescence methods were used for the measurement of osteocalcin, chemiluminescence for PTH, enzyme-linked immunosorbent assay (ELISA), for NTX and radioimmunoassay for intact PINP. Vitamin B6 and 25-D_3_ were measured by liquid chromatography with tandem mass spectrometry (LC-MS-MS). Selenium and zinc were measured by inductively-coupled plasma/mass spectrometry (ICP-MS). The tests were performed after a 12-h fast, and the urine sample was a morning spot urine. The volume of blood on each visit was 57.5 mL, and the samples were stored at −70 degrees Celsius until measurement.

### 2.5. Statistical Analysis

A sample size of 44 subjects per group was needed to provide 90% power to detect a difference of −110 ng/L in s-CTX-1, assuming an SD of 144 based on α report by Kruger et al. [7] and a two-tailed significance level of α = 0.025. To allow for a 20% dropout rate, 54 subjects (total 108) were randomised per treatment arm.

Similarly, 30 subjects per group were needed to provide 90% power to detect a 12.5 difference (test minus control group) in c-OC/uc-OC, assuming a standard deviation (SD) of 13 based on the study by Binkley et al. [14] and a two-tailed significance level of α = 0.025. To allow for a 20% dropout rate, a total of 36 participants (total = 72) were to be randomised per treatment arm. However, as the sample size for the first primary endpoint was larger, the overall sample size required for this study was 108 in total.

There were three analysis populations. Randomised subjects were all those subjects who were randomised and might or might not have received the study product. The safety population included all subjects who received at least one dose of the study product. Assessment of efficacy was based on the intent-to-treat (ITT) population, which included all subjects in the safety population who had at least one post product co-primary efficacy assessment (either s-CTX-1 or c-OC/uc-OC).

The co-primary efficacy variables of the study were the change from baseline in bone resorption marker (s-CTX-1) after 6 months and the change from baseline in c-OC/uc-OC after 6 months. The efficacy variables were analysed using analyses of covariance (ANCOVA) including the product group and age strata as fixed effects, and the corresponding baseline value as the covariate. Adjusted means, within-product *p*-values for each product group, product group difference expressed in percentage change from baseline, and the between-product *p*-values based on the statistical model described above were calculated. Statistical tests to compare treatments were two-sided and were employed at a level of significance of α = 0.025 for the co-primary endpoints. Assumptions of normality and homogeneity of variances in the ANCOVA model were evaluated after study unblinding. If violations were observed, then either suitable data transformations were performed as a post-hoc sensitivity analysis or a non-parametric Van Elteren test was used, and results were to be compared with the primary analysis results. To visually inspect the treatment effect of s-CTX-1 and c-OC/uc-OC, plots across time (baseline, month 3, and month 6) were displayed with least square (LS) means and ±SE bars, which were obtained from the ANCOVA analysis.

As a post-hoc sensitivity analysis, s-CTX-1 was log transformed (natural logarithm), and the log-transformed data were analysed using the ANCOVA model with the product group and age strata as fixed effects, and the corresponding log transformed baseline value as a covariate. The interpretation of the data was based on the geometric mean ratio, 95% confidence interval, and *p*-value.

A post-hoc subgroup ANCOVA analysis was performed on one of the co-primary variables, s-CTX-1 by age strata using the ITT population. The ANCOVA model had the product group, strata, and the product and strata interaction as fixed effects, and the baseline value of s-CTX-1 as the covariate. Statistical tests to compare treatments were two-sided and employed a level of significance of α = 0.05.

For the secondary efficacy analyses, a similar protocol was followed, using ANCOVA as described above. The level of significance used was α = 0.05. A post-hoc sensitivity analysis was performed using the following variables: urinary Ca/CR, vitamin D3 (using 25 OH D3), and plasma vitamin B6. These variables were log transformed (natural logarithm) and the log-transformed data were analysed using the ANCOVA model with product group and age strata as fixed effects, and the corresponding log transformed baseline value as a covariate. The interpretation of the data was based on the geometric mean ratio and *p*-value. The efficacy variable of serum folate was analysed using the non-parametric Van Elteren test stratified by age, and the interpretation of data was based on *p*-values obtained from this test. All analyses were performed in SAS 9.4 (SAS Institute Inc., Cary, NC, USA).

## 3. Results

### 3.1. Demographics

A total of 117 subjects were screened for entry into the study; 114 subjects were enrolled (*n* = 57 in each group), of which 102 (89.5%) completed the study (Figure 1).

Overall, the demographics and clinical characteristics were comparable between the groups (Table 2). The mean (SD) age was 34.6 (4.60) years, with a range from 25 to 44 years. Overall, 53 randomised subjects (46.5%) were in the age group of 25 to less than 35 years, and 61 randomised subjects (53.5%) were in the age group of 35 to 45 years. Subjects had a mean (SD) height of 153.4 (5.12) cm, body weight of 58.5 (7.89) kg, and BMI of 24.8 (2.90) kg/m^2^. The number (%) of subjects compliant with the consumption of study products in the test product group and the control group were 51 (98.1%) and 48 (96.0%), respectively.

### 3.2. Efficacy

#### 3.2.1. Primary Efficacy Results

The primary endpoints used were the change in s-CTX-1 and c-OC/uc-OC over 6 months (Table 3). In both groups, there was a significant decrease (considered to be associated with improved bone health) in s-CTX after 6 months of treatment (−33% at the test group, and −23% at the control group, *p* < 0.001). Although the decrease was more at the test group, i.e., favouring the test product, the between-group differences were not significant at the 2.5% level (*p*-value 0.132).

A post-hoc analysis to assess the potential impact of age on the treatment effect was performed. At month 6, the *p*-value for interaction between the product group and the age strata was not statistically significant (*p* = 0.193) at the 5% level. Another post-hoc analysis was performed as the underlying assumptions of normality and homogeneity of variance had a slight deviation from normality. This involved log-transforming the data. The between-group adjusted geometric mean ratio (test over the reference product) favoured the test product but was not statistically significant at the 2.5% level (*p* = 0.121). This is shown in Table 4.

There was a significant increase in c-OC/uc-OC after 6 months of treatment in the test group (+53%, *p* < 0.001). The increase from baseline at the control group was not significant (*p*-value 0.308). The between-group differences were not significant at the 2.5% level (*p*-value 0.047).

#### 3.2.2. Secondary Efficacy Results—Bone Metabolism Parameters

All the results that involve parametric tests are summarised in the Table 5. The adjusted mean (± standard error) plots by time and treatment for s-CTX-1, c-OC/uc-OC, s-PINP, and BSAP are shown in Figure 2. The results of the post-hoc analyses using log-transformed data and non-parametric tests are shown in Table 4.

The decrease in s-CTX-1 observed at 3 months in both groups was significant when compared to baseline, but the between-group differences were not significant (*p*-value 0.181). Once again, the post-hoc analysis to test the impact of treatment did not show a significant interaction between the product group and the age strata at month 3 (*p*-value 0.691). The log transformation post-hoc analysis resulted in a between-group adjusted geometric mean ratio (test over reference product), which favoured the test product, and it was statistically significant at the 5% level (*p* = 0.043). This is shown in Table 4.

Serum NTX-1, another bone resorption marker, showed a significant decrease at 6 months in both groups, but between-group differences were not significant (*p*-value 0.128).

In terms of bone formation markers, the increase observed in c-OC/uc-OC at 3 months favoured the test group (+67% vs. +23%, *p*-value: 0.018). There was a decrease in s-PINP in both groups at 3 and 6 months. The decrease was more pronounced, and thus more favourable, at the test group (at 6 months −21% vs. −9%, *p*-value: 0.007). BSAP decreased more at the test group at 3 months and 6 months (between-group differences at 3 and 6 months: *p*-values 0.004 and < 0.001, respectively).

When evaluating the calcium status, s-PTH increased significantly at both groups at 6 months, with a more pronounced and thus less favourable increase observed at the control group, but the between-group differences were not significant (*p*-value: 0.159). Serum calcium increased slightly more at 6 months at the test group (between-group difference *p*-value: 0.041), but both phosphorus and the urinary calcium to creatinine ratio did not show any significant differences between groups, except when the data for the urinary calcium to creatinine ratio were log transformed, there was a significant increase for the test group at 3 months (Table 4).

#### 3.2.3. Secondary Efficacy Results—Micronutrients

For selenium and zinc, the between-group differences at both 3 and 6 months were not significant (Table 6).

For serum folate, the underlying assumption of normality and homogeneity of variance was examined during analysis, and significant deviations from normality were observed. Normality was not achieved after log-transforming the data, so the non-parametric Van Elteren test was performed as a post-hoc analysis. The between-group difference was statistically significant at month 3 and at month 6 (*p*-value < 0.001). The median change from baseline, which is robust to outliers, is indicative of an increase in the test group at month 3 (+3.60 nmol/L or 19%) and at month 6 (+7.95 nmol/L or 42%).

A post-hoc analysis was also performed for vitamin B6, as there was a slight deviation from normality. At months 3 and 6, the between-group adjusted geometric mean ratio (test over reference product) favoured the reference product (both statistically significant with *p*-values < 0.0001).

Vitamin B12 did not show a significant difference between groups at months 6, but there was a significant difference at month 3, favouring the test product (Table 6).

### 3.3. Safety

Overall, both supplements were well tolerated. No deaths, SAEs, or treatment emergent adverse events (TEAEs) leading to premature discontinuation of the study product were reported in any group. All the reported TEAEs were mild in intensity. A total of 19 (33.3%) subjects in each group reported at least one TEAE (Table 7). Of these, 14 subjects from the test group and 12 subjects from the control group reported treatment-related TEAEs (17 TEAEs in each group).

The most frequently reported TEAEs were gastrointestinal disorders, which were more frequently reported in the test group, 11 subjects (19.3%) compared with seven subjects (12.3%) in the control group. The most common (>5% in any group) TEAE was nausea (test vs. control: 8.8% vs. 3.5%). Other infrequently occurring TEAEs were constipation, dyspepsia, abdominal distension, flatulence, abdominal pain, abdominal pain (upper), diarrhoea, and vomiting. There were no remarkable trends in haemoglobin, vital signs, and physical examination within or between treatment groups.

## 4. Discussion

### 4.1. Bone Turnover Markers and Bone Metabolism

The rationale for selecting CTX-I as the co-primary endpoint was that in older women, a high level of this marker is associated with an increase in fracture risk [19]. In addition, when an anti-resorptive drug is used in osteoporosis, the greater the reduction in CTX-I, the greater the reduction in the risk of fracture, particularly vertebral fracture [20]. The test product did cause a significant decrease in CTX-I of over 30%, and this is in line with other clinical trials of calcium supplementation in younger women. For example, in New Zealand women given 1000 mg daily for 16 weeks [7], CTX-I decreased by 39% and in British women given up to 676 mg daily for 4 weeks, CTX decreased by 20% [9]. The finding that was unusual in this study was that the control group also had a decrease in CTX, although at 3 months, this was less marked than in the test group. The administration of the extra calories (216 kCal daily) may have caused weight gain, and such changes can result in decreases in bone resorption [21]. An alternative explanation is that bone resorption markers show large variability, and so, this might explain these unexpected changes. The other bone resorption marker (serum NTX) showed a similar pattern of change to CTX.

The changes in bone formation markers were quite different to those in bone resorption markers. PINP is the bone formation marker recommended by the IOF/IFCC [6]. As for CTX, when an anti-resorptive drug is used in osteoporosis, the greater the reduction in PINP, the greater the reduction in the risk of fracture, particularly vertebral fracture [20]. The test product did cause a significant decrease in PINP of 20%, and this is in line with other clinical trials of calcium supplementation in younger women. For example, in New Zealand women given 1000 mg daily for 16 weeks [7], PINP decreased by about 20%. The changes in BSAP were slightly lower, being around 5% in this study. The changes in bone formation markers did differ between the test and control groups in contrast to bone resorption markers. They do tend to be more responsive to repletion with vitamin D than bone resorption markers. In this study, the baseline 25 hydroxy vitamin D (25-OHD) levels were on average well below the level of sufficiency (50 nmol/L) [16]. However, the supplement of vitamin D used in this study was vitamin D2, but the assay was for vitamin D_3_, so we cannot know the effect of the supplement on 25-OHD_2_ levels.

Supplementation with calcium and vitamin D might be expected to increase serum and urinary calcium and to decrease parathyroid hormone levels. There was a significant 3% increase in serum calcium at 6 months and a significant 52% increase in urinary calcium to creatinine ratio at 3 months (based on the ratio of the geometric means). The changes in PTH were unexpected, with increases in both groups. There is evidence from trials of calcium supplements with or without vitamin D that bone loss and fractures can be prevented in older people [10,22,23]

### 4.2. Vitamin K Status

The rationale for measuring the ratio of carboxylated to undercarboxylated OC was that it is a marker of vitamin K status. The mean level at baseline in both groups was higher than that reported by others, indicating some degree of deficiency of vitamin K [14,24,25]. Vitamin K is essential for the gamma-carboxylation of bone proteins, including those present in bone (such as OC, matrix Gla-protein, and protein S). As changes in uc-OC and c-OC are in opposite directions, their ratio reflects an estimate of vitamin K nutrition [12]. A clinical trial of a commonly used form of vitamin K2 (namely MK-7, at a dose of 375 mcg daily for 1 year) prevented bone loss, particularly trabecular bone [24]. Furthermore, a meta-analysis of randomised controlled trials indicated that vitamin K2 administration reduces uc-OC and increases c-OC, and it is associated with reduced bone loss and possibly a reduction in the risk of fractures [13]. The dose of 55 mcg of MK-7 given in this study had a significant benefit (increase) at 3 months but not at 6 months.

### 4.3. Other Micronutrients

Vitamins B_12_, B_6_, and folic acid are cofactors in homocysteine metabolism, and supplementation with B vitamins is effective in lowering homocysteine levels in humans [26]. High homocysteine levels have been associated with an increase in the risk of fracture, including hip fracture [27]. One large clinical trial did not show any effect of vitamin B supplementation on fracture risk [26].

Following 6 months’ supplementation with 2 mg vitamin B6, there was an adjusted mean increase of 9 nmol/L in the test group (resulting in a similar level to that found in US adults), indicating satisfactory supplementation. The mean baseline plasma vitamin B6 in the test group (23.6 nmol/L, Table 2) was below the mean level of 32 nmol/L among US adults (aged between 21 and 44 years) and close to the threshold for deficiency of 20 nmol/L [17].

The threshold for vitamin B12 deficiency is 148 pmol/L, and the mean baseline B12 in the test group was 177 pmol/L. Following 1 mcg daily supplementation (the UK recommends 1.5 mcg vitamin B12 daily [18]), mean serum vitamin B12 in the test group increased by 50 pmol/L at 6 months (an increase of 35%). There were significant increases in serum folate (Table 4) based on non-parametric analyses. Whether these changes in B vitamins would have an effect on fracture risk remains unknown.

## 5. Conclusions

This randomised controlled 6-month trial of a nutritional supplement showed favorable changes in bone turnover markers (decreased) and calcium homeostasis; such changes in older adults have been associated with slowing of bone loss and reduced fracture risk. There were also changes in the ratio of carboxylated to undercarboxylated osteocalcin (as a result in the improvement in vitamin K status), and such changes have been linked to slowing of bone loss in older subjects. Finally, there were improvements in the status of B vitamin, and such changes have been associated with reductions in homocysteine, but it is uncertain whether this would affect fracture risk. The product was generally well tolerated. This study shows the nutritional supplement holds promise for improved bone health among young Indian women. The next step may be a trial with bone mineral density as the endpoint.

## Figures and Tables

**Figure 1 nutrients-13-00364-f001:**
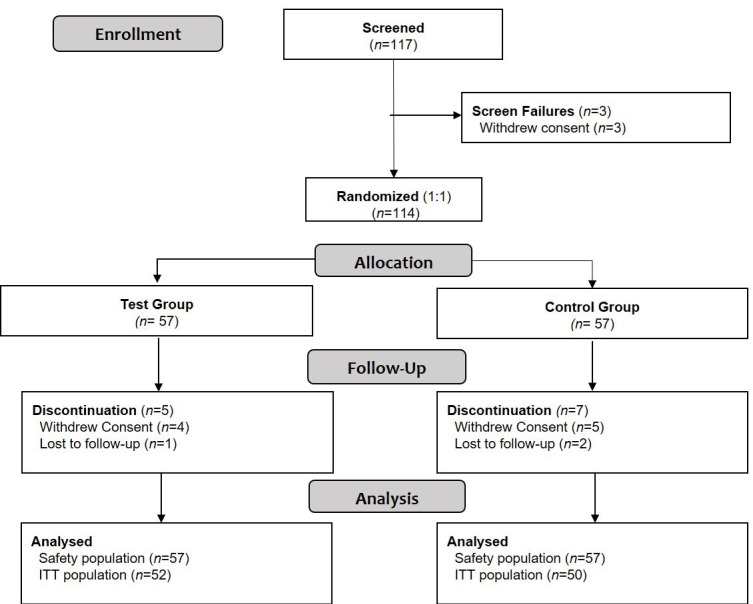
Consort diagram. ITT: intent-to-treat.

**Figure 2 nutrients-13-00364-f002:**
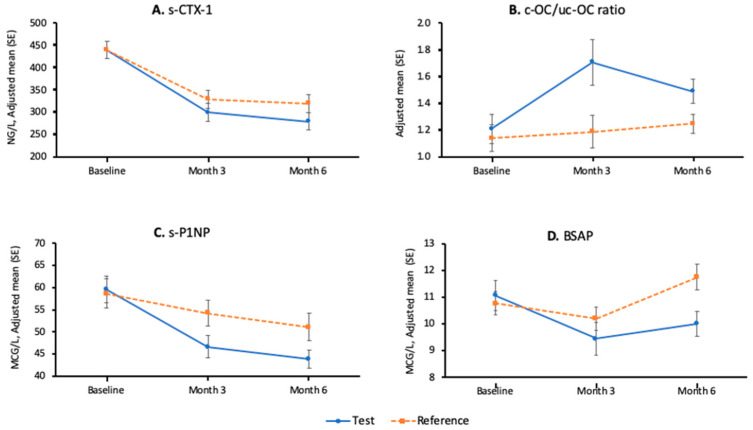
Adjusted mean (± standard error) plots by time and treatment (intent-to-treat (ITT)) population: *n* = 102. (**A**): s-CTX-1; (**B**): c-OC/uc-OC ratio; (**C**): s-PINP; (**D**): BSAP.

**Table 1 nutrients-13-00364-t001:** Composition of supplements. ICMR: Indian Council of Medical Research; RDA 2010: Recommended dietary allowances; WHO 2004: World Health Organisation; Kcal: kilocalorie.

Nutritional Composition		Nutritional Supplement (Test Group)	Placebo (Control Group)
	ICMR/WHO/RDA	Amount per daily serving (% RDA)	Amount per daily serving
Energy (Kcal)	1900	215.9 (11.4)	215.9
Carbohydrate (g) Of which sugar (g)	NA	40.9 0	47 10.8 (approximately)
Fat (g)	20	1.8 (9)	2 (max)
Protein (g)	55	9 (16.4)	3 (max)
Vitamin A (mcg)	600	198 (33)	
Vitamin D2 (IU)	400	400 (100)	
Vitamin K2 (mcg)	55	55 (100)	
Vitamin E (mg)	10	2.5 (25)	
Vitamin B1 (mg)	1	0.3 (33)	
Vitamin B2 (mg)	1.1	1.1 (100)	
Niacin (mg)	12	4 (33)	
Vitamin B6 (mg)	2	2 (100)	
Vitamin B12 (mcg)	1	1 (100)	
Folic acid (mcg)	120	120 (100)	
Vitamin C (mg)	40	40 (100)	
Biotin (mcg)	30	9.9 (33)	
Pantothenic acid (mg)	5	1.7 (34)	
Calcium (mg)	600	600 (100)	
Iodine (mcg)	150	49.5 (33)	
Magnesium (mg)	310	72.6 (23.4)	
Zinc (mg)	10	1.6 (16.2)	
Selenium (mcg)	40	25.8 (64.5)	
Potassium (mg)	3225	325 (10.1)	

**Table 2 nutrients-13-00364-t002:** Demographics, nutritional status, and baseline characteristics.

	Test	Control	Reference Interval
General Characteristics			
Female, *n* (%)	57 (100)	57 (100)	
Age, years [mean (SD)]	35.0 (4.4)	34.2 (4.8)	
Height, cm [mean (SD)]	152.8 (4.9)	154.1 (5.3)	
Weight (kg) [mean (SD)]	57.5 (7.1)	59.6 (8.5)	
Body mass index, kg/m^2^ [mean (SD)]	24.7 (3.0)	25.0 (2.8)	
Bone Metabolism Parameters, Mean (SD)			
Primary endpoints			
s-CTX-1 (ng/L)	435 (119)	440 (160)	111 to 791 [15]
c-OC/uc-OC	1.2 (0.7)	1.2 (0.8)	0.69 ± 0.41 [12]
Secondary endpoints			
CTX-I/CR, mcg/mmol	2.6 (1.6)	2.4 (1.3)	N/A
s-NTX-I (nM BCE/L)	11.5 (3.3)	11.9 (3.0)	Female 6.2 to 19
s-PINP (mcg/L)	59.6 (21.6)	58.0 (22.2)	17.3 to 75.2 [15]
Osteocalcin (mcg/L)	19.8 (5.2)	19.3 (5.5)	7.0 to 38.0
ALP (IU/L)	67.8 (25.1)	69.8 (17.9)	20 to 125
BSAP (mcg/L)	10.9 (4.0)	10.7 (2.9)	4.7 to 18.8
s-PTH (pmol/L)	5.7 (2.8)	5.4 (3.0)	1.1 to 6.8
Calcium (mmol/L)	2.30 (0.06)	2.30 (0.07)	2.12 to 2.56
Phosphorus (mmol/L)	1.15 (0.13)	1.13 (0.14)	0.8 to 1.45
Urinary Ca/CR (mmol/mol CR)	107.4(85.0)	107.7 (83.1)	28 to 905
25(OH)D3 (nmol/L)	43.2 (35.4)	41.6 (19.9)	>50 [16]
Micronutrients, Mean (SD)			
Selenium (mcmol/L)	1.34 (0.17)	1.31 (0.20)	0.83 to 2.03
Zinc (mcmol/L)	10.8 (1.7)	10.6 (2.0)	9.2 to 19.9
Folate (nmol/L)	18.5 (13.2)	36.5 (129.3)	>12.3
Vitamin B6 (nmol/L)	23.4 (15.5)	21.8 (13.3)	8 to 88
Vitamin B12 (pmol/L)	174.8 (72.5)	206.8 (132.3)	148 to 812

s-CTX-1: serum cross-linked C-telopeptide of type I collagen; c-OC/uc-OC: ratio of carboxylated to under-carboxylated osteocalcin; CTX1/CR: serum cross-linked C-telopeptide of type I collagen/creatinine ratio; s-NTX-1: serum N-terminal telopeptide of type I collagen; s-PINP: serum procollagen type I N propeptide; ALP: alkaline phosphatase; s-PTH: serum parathyroid hormone; Ca/Cr: calcium-to-creatinine ratio. The ranges were provided from the manufacturer unless stated otherwise.

**Table 3 nutrients-13-00364-t003:** Changes in biochemistry in response to test and control products over six months—primary endpoints.

		Test (*n* = 52)	Control (*n* = 50)	Difference
s-CTX-1 (ng/L)				
Baseline	Mean (SD)	444 (119)	441 (162)	
6 months	Mean (SD)	283 (142)	324 (168)	
	% Change from baseline, (SD)	−33 (41) ***	−23 (34) ***	NS
c-OC/uc-OC				
Baseline	Mean (SD)	1.21 (0.76)	1.14 (0.68)	
6 months	Mean (SD)	1.49 (0.67)	1.25 (0.49)	
	% Change from baseline, (SD)	53 (82) ***	45 (134)	NS

Analysis was performed using the analyses of covariance (ANCOVA) model with product group and age strata as fixed effects, and the corresponding baseline value as covariate. Data are presented as adjusted mean. Note that for CTX, a post-hoc analysis was performed as the underlying assumptions of normality, and homogeneity of variance had a slight deviation from normality. This involved log-transforming the data, and the results are shown in Table 4. s-CTX-1: serum cross-linked C-telopeptide of type I collagen; c-OC/uc-OC: ratio of carboxylated to under-carboxylated osteocalcin; SD: standard deviation. *p*-values: *** < 0.001; NS: not significant.

**Table 4 nutrients-13-00364-t004:** Post-hoc tests for primary and secondary efficacy results that had deviations from the assumption of normality.

	Geometric Mean Ratio (Test over Reference Product)	*p*-Value
s-CTX-1 (ng/L)		
3 months	0.84	**0.043**
6 months	0.85	0.121
Urinary Ca/Cr		
3 months	1.52	**0.014**
6 months	0.73	0.242
Vitamin D3		
3 months	0.84	**0.003**
6 months	0.90	0.308
Folate *		
3 months	NA	**<0.001**
6 months	NA	**<0.001**
Vitamin B6		
3 months	1.75	**<0.001**
6 months	1.60	**<0.001**

These variables were log transformed (natural logarithm), and the log-transformed data were analysed using the ANCOVA model with product group and age strata as fixed effects, and the corresponding log-transformed baseline value as a covariate. The interpretation of the data was based on the geometric mean ratio and *p*-value. * The efficacy variable of serum folate was analysed using the non-parametric Van Elteren test stratified by age, and the interpretation of data was based on *p*-values obtained from this test. s-CTX-1: serum cross-linked C-telopeptide of type I collagen; Urinary Ca/Cr: urinary calcium-to-creatinine ratio; NS: not significant; NA: not available. For s-CTX-1 at 6 months (primary efficacy result), the level of significance was at 2.5%, while for the other parameters, the level was at 5%. The *p*-values shown in bold were statistically significant.

**Table 5 nutrients-13-00364-t005:** Changes in bone metabolism parameters (secondary endpoints) in response to test and control products over six months.

		Test	Control	Difference
s-CTX-1 (ng/L)				
Baseline	Mean (SD)	444 (119)	441 (162)	
3 months	Mean (SD)	300 (158)	330 (167)	
	% Change from baseline, (SD)	−32 (37) ***	−24 (28) ***	NS
c-OC/uc-OC				
Baseline	Mean (SD)	1.21 (0.76)	1.14 (0.68)	
3 months	Mean (SD)	1.71 (1.21)	1.19 (0.83)	
	% Change from baseline, (SD)	67 (124) ***	23 (110)	*
s-NTX-1 (nM BCE)				
Baseline	Mean (SD)	11.6 (3.4)	11.9 (3.1)	
3 months	Mean (SD)	10.7 (6.3)	10.0 (5.7)	
	% Change from baseline, (SD)	−3.0 (58.5)	−14.1 (50.4) *	NS
6 months	Mean (SD)	8.2 (3.1)	9.4 (4.2)	
	% Change from baseline (SD)	−25.2 (33.3) ***	−19.0 (32.3) ***	NS
s-PINP (mcg/L)				
Baseline	Mean (SD)	59.6 (22.0)	58.7 (23.1)	
3 months	Mean (SD)	46.6 (18.0)	54.2 (21.0)	
	% Change from baseline, (SD)	−19 (22) ***	−4 (29) *	**
6 months	Mean (SD)	43.9 (14.2)	51.1 (23.0)	
	% Change from baseline, (SD)	−21 (25) ***	−9 (30) ***	**
ALP (IU/L)				
Baseline	Mean (SD)	69.0 (26.0)	70.4 (18.4)	
3 months	Mean (SD)	63.3 (23.4)	67.7 (17.7)	
	% Change from baseline, (SD)	−8 (11) ***	−2 (13)	*
6 months	Mean (SD)	63.4 (20.3)	70.2 (17.0)	
	% Change from baseline, (SD)	−5 (23) **	1 (14)	*
BSAP (mcg/L)				
Baseline	Mean (SD)	11.1 (4.1)	10.8 (3.1)	
3 months	Mean (SD)	9.4 (4.3)	10.2 (3.1)	
	% Change from baseline, (SD)	−16 (20) ***	−2 (38) *	**
6 months	Mean (SD)	10.0 (3.3)	11.8 (3.4)	
	% Change from baseline, (SD)	−5 (35) *	10 (13) *	***
s-PTH (pmol/L)				
Baseline	Mean (SD)	5.7 (2.9)	5.3 (2.5)	
3 months	Mean (SD)	5.0 (3.4)	5.8 (2.6)	
	% Change from baseline, (SD)	−6 (46)	16 (59)	NS
6 months	Mean (SD)	7.3 (3.4)	8.1 (4.3)	
	%Change from baseline, (SD)	43 (78) **	70 (106) ***	NS
Calcium (mmol/L)				
Baseline	Mean (SD)	2.30 (0.06)	2.30 (0.07)	
3 months	Mean (SD)	2.27 (0.08)	2.26 (0.09)	
	% Change from baseline, (SD)	−2 (4) **	−2 (4) ***	NS
6 months	Mean (SD)	2.40 (0.08)	2.33 (0.08)	
	% Change from baseline, (SD)	3 (4) ***	1 (4) **	*
Phosphorus (mmol/L)				
Baseline	Mean (SD)	1.14 (0.13)	1.13 (0.14)	
3 months	Mean (SD)	1.16 (0.19)	1.18 (0.16)	
	% Change from baseline, (SD)	3 (18)	6 (15) **	NS
6 months	Mean (SD)	1.13 (0.15)	1.12 (0.18)	
	% Change from baseline, (SD)	0 (15)	0 (16)	NS

Analysis was performed using the ANCOVA model with product group and strata as fixed effects and the corresponding baseline value as covariate. Data are presented as adjusted mean. Note that for CTX, a post-hoc analysis was performed as the underlying assumptions of normality and homogeneity of variance had a slight deviation from normality. This involved log-transforming the data, and the results are shown in Table 4. s-CTX-1: serum cross-linked C-telopeptide of type I collagen; c-OC/uc-OC: ratio of carboxylated to under-carboxylated osteocalcin; CTX1/CR: serum cross-linked C-telopeptide of type I collagen/creatinine ratio; s-NTX-1: serum N-terminal telopeptide of type I collagen; s-PINP: serum procollagen type I N propeptide; ALP: alkaline phosphatase; s-PTH: serum parathyroid hormone; Ca/Cr: calcium-to-creatinine ratio; SD: standard deviation; *p*-values: *** < 0.001; ** 0.001–0.010; * 0.010–0.050; NS: not significant.

**Table 6 nutrients-13-00364-t006:** Changes in micronutrients (secondary endpoints) in response to test and control products over six months.

		Test	Control	Difference
Selenium (mcmol/L)				
Baseline	Mean (SD)	1.35 (0.17)	1.31 (0.18)	
3 months	Mean (SD)	1.29 (0.22)	1.25 (0.29)	
	% Change from baseline, (SD)	−3 (19)	−4 (20) **	NS
6 months	Mean (SD)	1.41 (0.28)	1.35 (0.26)	
	% Change from baseline, (SD)	5 (21)	4 (22)	NS
Zinc (mcmol/L)				
Baseline	Mean (SD)	10.8 (1.7)	10.6 (2.1)	
3 months	Mean (SD)	10.3 (1.8)	10.0 (1.5)	
	% Change from baseline, (SD)	−3 (17)	−4 (21) **	NS
6 months	Mean (SD)	10.3 (1.3)	10.0 (1.6)	
	% Change from baseline, (SD)	−2 (17)	−4 (20) **	NS
Vitamin B12 (pmol/L)				
Baseline	Mean (SD)	177.1 (75.1)	209.6 (140.3)	
3 months	Mean (SD)	212.4 (96.7)	190.2 (77.0)	
	% Change from baseline, (SD)	22 (41) *	−1 (27)	*
6 months	Mean (SD)	227.4 (95.4)	205.4 (92.0)	
	% Change from baseline, (SD)	35 (51) **	6 (30)	NS

Analysis was performed using the ANCOVA model with product group and strata as fixed effects, and the corresponding baseline value as the covariate. Data are presented as adjusted mean. SD: standard deviation; *p*-values: ** 0.001–0.010; * 0.010–0.050; NS: not significant.

**Table 7 nutrients-13-00364-t007:** Summary of adverse events—Safety population.

Adverse Events Number (%) of Subjects [Events]	Test (*n* = 57)	Control (*n* = 57)
Number of subjects with at least one adverse event	19 (33.3) [17]	19 (33.3) [18]
Nausea	5 (8.8) [5]	2 (3.5) [2]
Constipation	1 (1.8) [1]	2 (3.5) [2]
Dyspepsia	1 (1.8) [1]	2 (3.5) [2]
Abdominal distension	1 (1.8) [1]	1 (1.8) [1]
Abdominal pain	2 (3.5) [2]	0
Flatulence	1 (1.8) [1]	1 (1.8) [1]
Abdominal pain upper	0	1 (1.8) [1]
Diarrhoea	0	1 (1.8) [1]
Vomiting	1 (1.8) [1]	0
Serum parathyroid hormone increased	2 (3.5) [2]	7 (12.3) [7]
Urine calcium/creatinine ratio increased	3 (5.3) [3]	6 (10.5) [6]
Vitamin D deficiency	7 (12.3) [7]	2 (3.5) [2]
Vitamin B12 deficiency	0	1 (1.8) [1]
Dysgeusia	0	1 (1.8) [1]
Headache	1 (1.8) [1]	0
Hunger	1 (1.8) [1]	0

## Data Availability

No data are provided.

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
