# Peer review of "Randomised Controlled Trial of Nutritional Supplement on Bone Turnover Markers in Indian Premenopausal Women"

_nutrients, 2021, doi:10.3390/nu13020364_

Round 1

Reviewer 1 Report

In their randomized controlled trial of Indian healthy premenopausal women, Umarji PB and colleagues investigated the short-term impact of nutritional supplementation on various bone markers. This is an important topic.

  1. A nutritional supplement was protein rich beverage powder fortified with multi-micronutrients including calcium (600 mg), vitamin D (400 IU) and vitamin K (55 mcg) per daily serving while a placebo supplement was low protein non-fortified isocaloric beverage powder. In the Abstract, more detailed information is helpful.
  2. In the Discussion, please note that the baseline levels of carboxylated/undercarboxylated osteocalcin would not suggest vitamin K deficiency. The ratio does not seem to be associated with bone mass (Cheung AM, et al. PLoS Med 2008;5:e196, Binkley N, et al. J Bone Miner Res 2009;24:983-991 and Ronn SH, et al. Osteoporos Int 2021;32:185-191), although vitamin K supplementation could reduce fracture risk.

Author Response

In their randomized controlled trial of Indian healthy premenopausal women, Umarji PB and colleagues investigated the short-term impact of nutritional supplementation on various bone markers. This is an important topic.

  1. A nutritional supplement was protein rich beverage powder fortified with multi-micronutrients including calcium (600 mg), vitamin D (400 IU) and vitamin K (55 mcg) per daily serving while a placebo supplement was low protein non-fortified isocaloric beverage powder. In the Abstract, more detailed information is helpful.

WE HAVE ADDED THIS NUTRITIONAL INFORMATION TO THE ABSTRACT.

  1. In the Discussion, please note that the baseline levels of carboxylated/undercarboxylated osteocalcin would not suggest vitamin K deficiency. The ratio does not seem to be associated with bone mass (Cheung AM, et al. PLoS Med 2008;5:e196, Binkley N, et al. J Bone Miner Res 2009;24:983-991 and Ronn SH, et al. Osteoporos Int 2021;32:185-191), although vitamin K supplementation could reduce fracture risk.

WE HAVE ADDED THE REFERENCES AS SUGGESTED AND DELETED THE SENTENCE

Reviewer 2 Report

The study confirmed the beneficial effect of calcium supplementation on bone turnover markers in premenopausal women that are at risk of undernutrition. Observed decreases of reference bone markers CTX and P1NP are well over the LSC in % of change from baseline  and in units of concentrations, as well. This proves that using supplementation may improve bone turnover leading to maintain good bone health in premenopausal women and avoid enhanced bone loss later in life. There are however several technical questions concerning laboratory methods used in the study. Details on blood and urine collections are missing. There is no info whether blood was collected in the fasting state (required for CTX assay) or postprandially (allowed for P1NP), there is no info how the urine collection was performed (was it the first or second morning void?). What was the volume of blood collected form each subject to perform so many assays with the use of different technologies? Were all assays performed in frozen/deep-frozen samples?

Surprisingly CTX and P1NP assays were performed with diffrent technologies while both are available on the same system. No information is given whether total or intact P1NP was measured.

The results for vitamin D are curious and difficult to understand. In Table 2 the values for 25(OH)D are presented, in Table 4 the data for D3 are given which may lead to confusion. It is widely accepted that total 25(OH)D reflects the status of this metabolite in humans but in submitted study only 25(OH)D3 was measured with LC/MS-MS which allows measurements of different vitamin D metabolites, also 25(OH)D2. Even more surprising is why total 25(OH)D was not measured in the same system as CTX.

What was the reason to measure both,  total ALP acivity and bone-specific  alkaline phosphatase?

The explanation of CTX decrease in the control group (line 354-356) is not convincing as this may probably be partly due to biological variation which for CTX accounts for up to 10-15%.

The calculated power of the test for changes in CTX and P1NP make the reader to believe that sample sizes are not too small.

Minor comments :

1.The study and control groups seem to be rather small however, the calculated  power of the test to detect differences in bone markers makes us to believe the sample sizes were sufficient.

Author Response

The study confirmed the beneficial effect of calcium supplementation on bone turnover markers in premenopausal women that are at risk of undernutrition. Observed decreases of reference bone markers CTX and P1NP are well over the LSC in % of change from baseline  and in units of concentrations, as well. This proves that using supplementation may improve bone turnover leading to maintain good bone health in premenopausal women and avoid enhanced bone loss later in life. There are however several technical questions concerning laboratory methods used in the study. Details on blood and urine collections are missing. There is no info whether blood was collected in the fasting state (required for CTX assay) or postprandially (allowed for P1NP), there is no info how the urine collection was performed (was it the first or second morning void?). What was the volume of blood collected form each subject to perform so many assays with the use of different technologies? Were all assays performed in frozen/deep-frozen samples?

YES, THEY HAD A 12-HOUR FAST AND THIS IS NOW ADDED.

THE URINE SAMPLE WAS A MORNING ‘SPOT’ URINE.

THE VOLUME OF BLOOD ON EACH VISIT WAS 57.5 ML.

THE SAMPLES WERE STORED AT -70 DEGREES C UNTIL MEASUREMENT.

Surprisingly CTX and P1NP assays were performed with different technologies while both are available on the same system. No information is given whether total or intact P1NP was measured.

IT WAS INTACT PINP. WE AGREE THAT PINP COULD HAD BEEN MEASURED BY THE ROCHE ASSAY, BUT THE STANDARD METHOD OFFERED BY THE COMMISSIONED LABORATORY WAS RIA

The results for vitamin D are curious and difficult to understand. In Table 2 the values for 25(OH)D are presented, in Table 4 the data for D3 are given which may lead to confusion. It is widely accepted that total 25(OH)D reflects the status of this metabolite in humans but in submitted study only 25(OH)D3 was measured with LC/MS-MS which allows measurements of different vitamin D metabolites, also 25(OH)D2. Even more surprising is why total 25(OH)D was not measured in the same system as CTX.

WE HAVE CHANGED 25(OH)D IN TABLE 2 IN ORDER TO CLARIFY THAT IT IS 25(OH)D3.

WE AGREE THAT TOTAL 25(OH)D COULD HAVE BEEN MEASURED BY CHEMOLUMINESCENCE. HOWEVER, THE STANDARD METHOD OFFERED BY THE COMMISSIONED LABORATORY IS LS/MS-MS.

What was the reason to measure both,  total ALP activity and bone-specific  alkaline phosphatase

BONE ALP IS A MARKER THAT IS SPECIFIC TO BONE AND SO IT WOULD BE MORE SENSITIVE THAN TOTAL ALP. THE TOTAL ALP IS A MORE WIDELY USED MEASUREMENT AND SO IT WAS INCLUDED FOR INTEREST.

The explanation of CTX decrease in the control group (line 354-356) is not convincing as this may probably be partly due to biological variation which for CTX accounts for up to 10-15%.

THANKS FOR THIS COMMENT. WE HAVE NOW ADDED A FURTHER COMMENT TO INTERPRETATION TO SAY, ‘AN ALTERNATIVE EXPLANATION IS THAT BONE RESORTION MARKERS SHOW LARGE VARIABILITY AND SO THIS MIGHT EXPLAIN THESE UNEXPECTED CHANGES’.

The calculated power of the test for changes in CTX and P1NP make the reader to believe that sample sizes are not too small.

THE ONLY PART OF THE POWER CALCULATION THAT CAN BE CHALLENGED (SINCE THIS IS DONE IN ADVANCE) IS THE STANDARD DEVIATION. WE STATED THAT IT WOULD LIKELY BE 140 NG/L. BASED IN TABLE 2, IT WAS 119 FOR THE TEST GROUP AND 160 FOR THE CONTROL GROUP.

Minor comments :

1.The study and control groups seem to be rather small however, the calculated  power of the test to detect differences in bone markers makes us to believe the sample sizes were sufficient.

THANK YOU FOR YOUR COMMENT.